



# Imbalanced phosphorus and nitrogen deposition in China's forests

Enzai Du[1], Wim de Vries[2,3], Wenxuan Han[4], Xuejun Liu[4], Zhengbing Yan[5], Yuan Jiang[1]

[1]State Key Laboratory of Earth Surface Processes and Resource Ecology, and College of Resources Science & Technology, Beijing Normal University, Xinjiekouwai Street 19#, Beijing, 100875, China

[2]Environmental Systems Analysis Group, Wageningen University, PO Box 47, 6700 AA Wageningen, The Netherlands

[3]Alterra, Wageningen University and Research Center, PO Box 47, 6700 AA Wageningen, the Netherlands

[4]College of Resources and Environmental Sciences, China Agricultural University, Beijing, 100193, China

[5]Department of Ecology, and Key Laboratory for Earth Surface Processes of the Ministry of Education, Peking University, Beijing, 100871, China

*Correspondence to*: Enzai Du (enzaidu@bnu.edu.cn) and Yuan Jiang (jiangy@bnu.edu.cn)

**Abstract.** Acceleration of anthropogenic emissions in China has substantially increased nitrogen (N) deposition during last three decades and may result in an imbalance of atmospheric N and phosphorus (P) inputs in terrestrial ecosystems. However, the status of P deposition in China is poorly understood. This study synthesized data on total P and total N concentrations in bulk precipitation and throughfall from published literature to assess the characteristics of P deposition, N deposition and N:P deposition ratio in China's forests. Our results show relatively high mean rates of atmospheric P inputs by bulk precipitation (0.38 kg P ha$^{-1}$ yr$^{-1}$) and throughfall (0.84 kg P ha$^{-1}$ yr$^{-1}$), but they were accompanied by even much higher N inputs by bulk precipitation (16.5 kg N ha$^{-1}$ yr$^{-1}$) and throughfall (26.2 kg N ha$^{-1}$ yr$^{-1}$), resulting in high N:P ratios in bulk precipitation (44.4) and throughfall (32.8), respectively. Total P and N concentrations both were significantly enriched in throughfall versus bulk precipitation, leading to an estimate of canopy captured dry deposition of 0.46 kg P ha$^{-1}$ yr$^{-1}$ and at least 9.7 kg N ha$^{-1}$ yr$^{-1}$, respectively. We found significantly higher P deposition and lower N:P ratios nearby than far from semiarid regions. Moreover, fluxes of total P and total N in bulk precipitation and throughfall both showed a significant power-law increase with closer distance to the nearest large cities. Our results suggest an anthropogenic alternation of regional P and N cycling, which might shift large areas of China's forests towards human-induced P limitation especially in southern China.

## 1 Introduction

Nitrogen (N) and phosphorus (P) are essential macronutrients, both of which widely limit primary productivity across terrestrial ecosystems (Elser et al., 2007; Vitousek et al., 2010). New N inputs to terrestrial ecosystems are driven by biological N fixation and atmospheric deposition, the latter being increasingly important (Cleveland et al., 2013). Acceleration of anthropogenic N emissions has substantially increased N deposition in China during last three decades and aroused widespread concerns about the consequent impacts on various ecosystems (Liu et al., 2011 & 2013; Cui et al., 2013). Enhanced N deposition often stimulates forest growth and hence carbon sequestration (Högberg, 2007; De Vries et al., 2009;



Thomas et al., 2010), but the expected growth acceleration might be diminished when the accompanied P supply is deficient (Braun et al., 2010; Li et al., 2016). Soil P availability in terrestrial ecosystems is primarily driven by mineral weathering and atmospheric deposition (Vitousek et al., 2010; Cleveland et al., 2013). Newman (1995) reviewed P deposition and weathering in global terrestrial ecosystems and estimated a range of $0.07 – 1.7$ kg P ha$^{-1}$ yr$^{-1}$ for P deposition and $0.01 – 1.0$ kg P ha$^{-1}$ yr$^{-1}$ for P weathering, indicating that both fluxes are in the same order of magnitude. More recent data on observed and modelled P deposition rates on a global scale, as presented by Mahowald et al. (2008), show a range of $0.01 − 3.0$ kg P ha$^{-1}$ yr$^{-1}$. Despite the significant role of P deposition, its status and characteristics are poorly understood in China.

Considering the importance of atmospheric deposition for N and P availability, the variation in N and P deposition may make a difference in N or P-limited ecosystems (Peñuelas et al., 2013). Spatial patterns and temporal trends of N deposition have recently been well characterized in China (Liu et al., 2013; Du and Liu, 2014; Lu and Tian, 2014; Jia et al., 2014; Du et al., 2014; Xu et al., 2015; Zhu et al., 2015). Most assessments on N deposition in China have been based on measurements of bulk deposition (Liu et al., 2013), which substantially underestimates the levels of total deposition because it consists mainly of wet deposition (Kulshrestha et al., 1995; Staelens et al., 2005; Chantara and Chunsuk, 2008). Throughfall deposition has been used as a more precise estimate of total deposition (Du et al., 2014 & 2015), although it is still an underestimate due to canopy N uptake, especially at lower N deposition levels (Draaijers et al., 1996; Sparks, 2009). A recent assessment has shown that mean throughfall deposition of inorganic N in China's forests was approximately 21.5 kg N ha$^{-1}$ yr$^{-1}$ (Du et al., 2014), which exceeds the critical loads of N deposition ($10 – 20$ kg N ha$^{-1}$ yr$^{-1}$) in most forest ecosystems (Bobbink et al., 2010). Unlike N, there is a lack of evaluation on P deposition across China. In the context of low soil P contents in terrestrial ecosystems across most areas of China (Jiang et al., 1986; Han et al., 2005), the role of P deposition is relevant because high-level N deposition, initially increasing forest growth, is likely shifting large areas of China's forests towards P limitation.

Previous studies have indicated that large cities in China are hotspots of N deposition (Du et al., 2014 & 2015; Jia et al., 2014) because the enhancement of N deposition is primarily driven by anthropogenic emissions from power generation, motor traffic and agricultural activities near urban areas (Liu et al., 2013; Jia et al., 2014). Unlike reactive N which can form stable gaseous compounds (e.g. $NO_x$ and $NH_3$), atmospheric P occurs primarily in the form of coarse particles and is dominantly derived from mineral dust in natural conditions (Mahowald et al. 2008; Vet et al., 2014). Higher P deposition is thus expected nearby the semiarid regions, where receive significant P-containing dust from neighbouring deserts or barren lands due to the effect of wind erosion (Okin et al., 2004; Mahowald et al. 2008). Based on measurements of P content of various fuels and estimates of the partitioning of P during combustion processes, a recent assessment indicates that anthropogenic sources, such as combustion-related emissions, can contribute to over 50% of the global atmospheric P and make a substantial contribution to global P deposition (Wang et al., 2015). We can thus expect higher P deposition nearby large cities with intensive anthropogenic P emissions, although the transportation of P-containing dust may lead to distinct urban hotspots of P deposition nearby and far from the semiarid regions. Here we synthesized data on total P and total N concentrations in bulk precipitation and throughfall from published literature to assess the status and characteristics of P deposition, N deposition and N:P deposition ratios in China's forests. More specifically, we addressed the following





questions: (1) What are the characteristics for concentrations and fluxes of P, N and N:P ratios in bulk precipitation and throughfall in China's forests? (2) How much of dry deposition of P and N is captured by forest canopies? (3) Do the concentrations and fluxes of P in bulk precipitation and throughfall increase similarly as N with closer distance to large cities?

## 2. Data and methods

### 2.1 Data sets

We collected data from published literatures on total P and total N concentrations in bulk precipitation and throughfall for typical forests in China, as well as information on site location (latitude and longitude) and annual precipitation. Total P and total N concentrations were assessed after digestion and include dissolved inorganic forms, dissolved organic forms and particulate forms. Observational data were selected only when precipitation and throughfall were measured simultaneously. Data on P and N concentrations for each sampling period were either taken directly from tables or digitized from figures using a GetData Graph Digitizer (Version 2.25, http://www.getdata-graph-digitizer.com). If concentration data at one site were measured for more than one forest stand or available for more than one year, a volume-weighted mean was calculated and used for further analysis. The distance between the sampling site and the nearest large city (non-agricultural population > 0.5 million) was derived using Google Earth for Microsoft Windows (Version 7.1.5.1557, Google Inc., USA).

Our database consisted of 33 sites which were distributed across main forest biomes in China (see Fig. S1 and Table S1 for detailed information). Annual precipitation ranged from 500 mm to 2900 mm. The distance from the sampling sites to the nearest large cities ranged from 5 km to 465 km. Using a criteria of a distance of 400 km to the semiarid regions (e.g. grassland), we grouped our datasets into sites nearby semiarid regions (n= 11) and sites far from the semiarid regions (n=22) (Fig. S1).

### 2.2 Statistical analysis

Phosphorus and N fluxes in the bulk precipitation and throughfall were estimated according to the volume-weighted mean concentration and annual precipitation. Canopy captured dry deposition was estimated as the difference between throughfall deposition and bulk deposition. The ratio of throughfall deposition versus bulk deposition was used to indicate the change in P or N fluxes through forest canopy. A Shapiro-Wilk normality test was used to test whether data were normally distributed. Data ranges were indicated by the first and third quantile. We used a paired sample t-test to test the differences of concentrations, fluxes and N:P ratios between bulk precipitation and throughfall. A Student's t-test was used to test the difference of N deposition levels between sites nearby the semiarid regions (n=11) and far from the semiarid regions (n=22).

Previous studies have indicated that N deposition shows a power-law increase with closer distance to the nearest large cities (Du et al., 2014 & 2015). To test the urban hotspot hypothesis of P deposition, we also used a power-law model to explore changes in concentrations and fluxes of total P with the distance between the sampling site and the nearest large city. The analysis was conducted separately for sites nearby the semiarid regions (n=11) and far from the semiarid regions (n=22).





Spatial patterns of P deposition, N deposition and N:P ratios were illustrated using ArcGIS Desktop (version 9.3, ESRI, USA). All statistical analysis was performed using R software (version 3.1.0; R Development Core Team, 2014, http://www.r-project.org/) with a significance level of $p < 0.05$.

## 3. Results

### 3.1 Phosphorus concentrations and fluxes in bulk precipitation and throughfall

Concentrations and fluxes of total P in bulk precipitation and throughfall were characterized by a log-normal distribution (Shapiro-Wilk normality test, $p > 0.10$). Geometric means of P concentration and flux in bulk precipitation were 0.031 (0.017 – 0.050) mg P $L^{-1}$ and 0.38 (0.21 – 0.57) kg P $ha^{-1}$ $yr^{-1}$, respectively. In throughfall, geometric means of P concentration and flux increased to 0.068 (0.041 – 0.138) mg P $L^{-1}$ and 0.84 (0.49–1.59) kg P $ha^{-1}$ $yr^{-1}$, respectively. Phosphorus concentrations were significantly enriched in throughfall versus bulk precipitation (paired sample t-test, $df = 32$, $p < 0.01$) and throughfall P deposition was on average 2.18 (1.40 – 3.11) times of bulk P deposition. The canopy captured dry P deposition, estimated as the difference between throughfall deposition and bulk deposition, was estimated at 0.46 (0.11 – 1.11) kg P $ha^{-1}$ $yr^{-1}$.

Spatial patterns of P deposition via bulk precipitation and throughfall showed large heterogeneity (Fig. 1a and 1b). Specifically, geometric means of P flux in bulk precipitation and throughfall were estimated to be 0.54 (0.39 – 0.98) and 1.22 (0.97 – 1.61) kg P $ha^{-1}$ $yr^{-1}$ at the 11 sites nearby the semiarid regions. At the other 22 sites far from the semiarid regions, geometric means of P flux in bulk precipitation and throughfall were significantly lower (Student's t-test, $df = 31$, $p < 0.01$), being 0.31 (0.20 – 0.50) and 0.68 (0.31 – 1.25) kg P $ha^{-1}$ $yr^{-1}$, respectively.

### 3.2 Nitrogen concentrations and fluxes in bulk precipitation and throughfall

Concentrations and fluxes of total N in bulk precipitation and throughfall were also log-normally distributed (Shapiro-Wilk normality test, $p > 0.10$). Geometric means of total N concentration and flux in bulk precipitation were 1.34 (0.88 – 2.19) mg N $L^{-1}$ and 16.5 (9.9 – 24.2) kg N $ha^{-1}$ $yr^{-1}$, respectively. In throughfall, geometric means of total N concentration and flux increased to 2.13 (1.23 – 3.87) mg N $L^{-1}$ and 26.2 (18.4 – 37.2) kg N $ha^{-1}$ $yr^{-1}$, respectively. Nitrogen concentrations were significantly enriched in throughfall versus bulk precipitation (paired sample t-test, $df = 31$, $p < 0.01$), leading to an estimate of canopy captured dry N deposition at 9.7 (2.9 – 15.7) kg N $ha^{-1}$ $yr^{-1}$. The ratio of throughfall deposition versus bulk deposition was 1.59 (1.20 – 1.96) for N, which was significantly lower than that for P (paired sample t-test, $df = 31$, $p < 0.01$). Spatial patterns of total N fluxes in bulk precipitation and throughfall showed several regional hotspots near large city clusters in central, eastern and southern China (Fig. 2a and 2b).





### 3.3 N:P ratios in bulk precipitation and throughfall

The N:P ratios in bulk precipitation and throughfall were also characterized by a log-normal distribution (Shapiro-Wilk normality test, $p > 0.10$). The geometric mean of N:P ratio was 44.4 (23.1 – 97.6) in bulk precipitation and decreased to 32.8 (18.7 – 63.6) in throughfall. At the 11 sites nearby the semiarid regions, the geometric mean of N:P ratio in bulk precipitation and throughfall were estimated at 19.3 (9.1 – 39.9) and 16.2 (10.9 – 28.6), respectively. Generally, the N:P ratios in bulk precipitation (geometric mean = 64.0, ranging 40.9 – 116.7) and throughfall (geometric mean =45.3, ranging 25.8 – 74.6) were significantly higher (Student's t-test, $df = 31$, $p < 0.01$) at the other 22 sites far from the semiarid regions, most of which were located in southern China (Figs. 3a and 3b).

### 3.4 Urban hotspots of phosphorus and nitrogen deposition

Fluxes of total P in bulk deposition and throughfall both showed a significant power-law increase with closer distance to the nearest large cities, either nearby or far from the semiarid regions (Figs. 4a, 4d, 5a & 5d). In line with the urban hotspot hypothesis, fluxes of total N in bulk precipitation and throughfall also showed a significant power-law increase with closer distance to the nearest large cities (Figs. 4b, 4e, 5b & 5e). At sites far from semiarid regions, N:P ratios in bulk precipitation and throughfall both showed no significant trend with changing distance to the nearest large cities (Figs. 4c & 4f). However, the N:P ratio in throughfall at sites nearby semiarid regions showed a significant power-law increase with closer distance to the nearest large cities, while no such trend was found for N:P ratio in bulk precipitation (Figs. 5c & 5f).

## 4. Discussion

### 4.1 Phosphorus deposition in China's forests

Atmospheric P-containing aerosols are either scavenged by precipitation or gravitationally deposited to the ground surface during dry weather. Based on very limited measurements of wet deposition around the world, Vet et al. (2014) summarized that total P fluxes in wet deposition range from 0.04 to 0.32 kg P ha$^{-1}$ yr$^{-1}$. However, most reported measurements are based on bulk deposition, which includes wet deposition plus a fraction of dry deposition. A recent synthesis of global datasets with 246 sites indicates that bulk deposition of total P is log-normally distributed and showed a geometric mean of 0.27 kg P ha$^{-1}$ yr$^{-1}$ (Tipping et al., 2014). However, large uncertainties remain in regional estimates due to the scarcity of observational data and unevenness of site distribution. For instance, mean bulk deposition of total P was estimated at 0.12 kg P ha$^{-1}$ yr$^{-1}$ in Asia based on measurements from only 7 sites (Tipping et al., 2014). Using observed data from 33 sites in China's forests, here we show much higher bulk deposition of total P with a geometric mean of 0.38 kg P ha$^{-1}$ yr$^{-1}$. Overall, our estimate of bulk P deposition in China is higher than the estimates for Europe (0.22 kg P ha$^{-1}$ yr$^{-1}$), Oceania (0.24 kg P ha$^{-1}$ yr$^{-1}$) and North America (0.29 kg P ha$^{-1}$ yr$^{-1}$) by Tipping et al. (2014), while it is lower than those in South America (0.43 kg P ha$^{-1}$ yr$^{-1}$) and Africa (0.62 kg P ha$^{-1}$ yr$^{-1}$).



Measurements of dry P deposition are rather scarce. Based on limited data of aerosol concentrations and the referential dry deposition velocity, Vet et al. (2014) estimated that annual dry deposition of total P on land ranges from 0.01 to 0.62 kg P ha$^{-1}$ yr$^{-1}$. Based on the P enrichment in throughfall versus bulk precipitation, the mean canopy captured dry P deposition was estimated as high as 0.46 kg P ha$^{-1}$ yr$^{-1}$ in China's forests, being significantly higher than bulk deposition (0.38 kg P ha$^{-1}$ yr$^{-1}$) (paired t-test, $df = 32$, $p < 0.01$). This result implies that dry deposition is the dominant pathway of atmospheric P deposition in China. Our assessment indicates that the geometric mean of throughfall P deposition in China's forests was 0.84 (0.49–1.59) kg P ha$^{-1}$ yr$^{-1}$, being in the global range of 0.01 − 3.0 kg P ha$^{-1}$ yr$^{-1}$ based on observed and modelled P deposition rates on a global scale (Mahowald et al., 2008). Using the mean throughfall deposition of 0.84 kg P ha$^{-1}$ yr$^{-1}$ and a forested area of 1.76 ×10$^8$ ha (Zhang et al., 2010), we further estimated a total P deposition of 0.15 Tg P yr$^{-1}$ in China's forests.

Previous assessments have suggested that mineral dust from neighbouring deserts contributes substantially to atmospheric P deposition in semiarid regions (Okin et al., 2004; Mahowald et al., 2008). Accordingly, our results also show that P deposition tended to be higher nearby the semiarid regions (Fig. 1a & 1b). Geometric means of P flux in bulk precipitation and throughfall were estimated to be as high as 0.54 (0.39 − 0.98) and 1.22 (0.97 − 1.61) kg P ha$^{-1}$ yr$^{-1}$ nearby the semiarid regions, while the values were significantly lower (Student's t-test, $df = 31$, $p < 0.01$) far from the semiarid regions, being 0.31 (0.20 − 0.50) and 0.68 (0.31 − 1.25) kg P ha$^{-1}$ yr$^{-1}$, respectively. This indicates a significant contribution of dust-borne sources to atmospheric P deposition in forest ecosystems nearby the semiarid regions.

**4.2 Nitrogen deposition in China's forests**

Our results show high levels of total N fluxes in bulk precipitation and throughfall in China's forests, on average being 16.5 kg N ha$^{-1}$ yr$^{-1}$ and 26.2 kg N ha$^{-1}$ yr$^{-1}$, respectively. Most previous assessments on N deposition in China were either based on bulk deposition or wet deposition (Liu et al., 2013; Du and Liu, 2014; Jia et al., 2014; Du et al., 2014; Zhu et al., 2015; Xu et al., 2015), showing similar values (13.7 to 19.3 kg N ha$^{-1}$ yr$^{-1}$) as our estimates based on bulk deposition in China's forests. Although throughfall may still underestimate total deposition due to canopy uptake (Draaijers et al., 1996; Sparks, 2009), it is a better proxy of total N deposition than bulk deposition because bulk deposition only accounts for 63% of throughfall deposition. In addition, the difference between throughfall deposition and bulk deposition leads to an estimate of canopy captured dry deposition of total N at 9.7 kg ha$^{-1}$ yr$^{-1}$, which is equivalent to 59 % of bulk deposition. Based on airborne concentration measurements via a nationwide N deposition monitoring network and inferential models, a more recent estimate indicated that average dry N deposition even exceeded wet/bulk deposition across China (Xu et al., 2015).

By integrating site-monitoring data of wet deposition and modelling results of dry deposition, Lu and Tian (2014) estimated that national mean of total inorganic N deposition was 20.1 kg N ha$^{-1}$ yr$^{-1}$. However, their assessment did not include organic N, which has been indicated to make a substantial contribution to total N in atmospheric deposition (Zhang et al., 2012; Du and Liu, 2014; Jickells et al., 2013). Compared with inorganic N fluxes in bulk precipitation (14.0 kg N ha$^{-1}$ yr$^{-1}$) and throughfall (21.5 kg N ha$^{-1}$ yr$^{-1}$) in China's forests (Du et al., 2014), we estimated that organic N on average



contributed to 2.5 and 4.7 kg N ha$^{-1}$ yr$^{-1}$ via bulk deposition and throughfall, respectively. Based on the mean throughfall deposition of 26.2 kg N ha$^{-1}$ yr$^{-1}$ and a forested area of $1.76 \times 10^8$ ha (Zhang et al., 2010), total N inputs via atmospheric deposition was thus estimated to be 4.6 Tg N yr$^{-1}$ in China's forests.

### 4.3 Urban hotspots of phosphorus and nitrogen deposition

Fluxes of total P in bulk deposition and throughfall both showed a significant power-law increase with closer distance to the nearest large cities, either far from (Figs. 4a, 4d) or nearby (Figs. 5a & 5d) the semiarid regions. Anthropogenic sources have been traditionally thought to make only a small contribution to P deposition (Okin et al., 2004; Mahowald et al., 2008). In contrast, our results emphasize that anthropogenic sources near large cities can have significant impacts on the spatial patterns of regional P deposition. The urban hotpot of P deposition might be derived from intensive combustion-related emissions near urban areas (Wang et al., 2015) and a short-distance transfer of P-containing aerosols from P-rich farmland soils (Anderson et al., 2006).

Spatial patterns of total N fluxes in bulk precipitation and throughfall showed several regional hotspots near large city clusters in central, eastern and southern China (Fig. 2a and 2b). In line with our previous assessments on inorganic N deposition (Du et al., 2014 & 2015), total N fluxes in bulk precipitation and throughfall were also in accordance to the urban hotspot hypothesis (Figs. 4b, 4e, 5b & 5e). This spatial pattern has been attributed to intensive motor traffic, energy production, waste treatment and agricultural activities (mainly N-fertilizer application and livestock breeding) near urban areas (Du et al., 2014 & 2015). Overall, our results suggest that rapid urbanization in China may have exerted significant alternation of regional P and N cycling by enlarging anthropogenic emissions and deposition. In addition, the urban hotspot model is an important approach to describe the way in which large cities shape the spatial pattern of P and N deposition. Therefore, it should be incorporated in modelling of P and N deposition at regional scales.

At sites far from semiarid regions, N:P ratios in bulk precipitation and throughfall both showed no significant trend with changing distance to the nearest large cities (Figs. 4c & 4f), indicating an synchronous changes in P and N deposition around these urban hotspots. However, N:P ratios in throughfall at sites nearby semiarid regions showed a significant power-law increase with closer distance to the nearest large cities, while no such trend was found for N:P ratio in bulk precipitation (Figs. 5c & 5f). This distinct trend of N:P ratios in bulk precipitation and throughfall suggest a more rapid increase in dry N deposition than in dry P deposition with closer distance to the nearest large cities.

### 4.4 Implications of imbalanced phosphorus and nitrogen deposition

Our results indicated an imbalance of N and P supply by atmospheric nutrient deposition in China's forests. The N:P ratio generally showed high values (geometric mean = 44.4) in bulk precipitation in China's forests. Although the N:P ratio decreased in the throughfall, the geometric mean (32.8) was still more than twice of that in tree leaves (geometric mean N:P ratio = 15) (Han et al., 2005) and three times of that in current-year twigs (geometric mean N:P ratio = 10.6) (Yao et al., 2015). It is also much higher than critical N:P ratios related to relative P limitation in view of forest growth, which are near





15 for coniferous forests and near 25 for deciduous forests (after Mellert and Göttlein, 2012). Compared to the sites nearby semiarid regions, the imbalance of P and N deposition was more intense in forests of southern China, showing significantly higher N:P ratios in bulk precipitation (geometric mean = 64.0, ranging 40.9 – 116.7) and throughfall (geometric mean =45.3, ranging 25.8 – 74.6) (Figs. 3a & 3b). Moreover, the ratio of total N versus total P may overestimate the relative P supply versus N because a fraction of atmospheric N deposition is most likely already taken up by the canopy (Draaijers et al., 1996; Sparks, 2009) and a fraction of atmospheric P usually is not bioavailable (Mahowald et al., 2008; Tipping et al., 2014).

The imbalance of N and P deposition may lead to an increase in the N:P ratio of soils and plant tissues and hence a shift towards human-induced P limitation (Peñuelas et al., 2013). Accordingly, a recent assessment has shown that foliar N concentration of woody plants in China's non-agricultural ecosystems increased significantly between the 1980s and the 2000s, while leaf P concentration did not change significantly over the same period (Liu et al., 2013). In addition, mean leaf P concentrations of China's plants were found to be significantly lower than the global averages, most likely due to lower soil P content (Han et al., 2005). Lower leaf P may constrain the response of photosynthetic capacity to leaf N as P-deficient plants have limited ribulose-1,5-bisphosphate regeneration (Reich et al., 2009). Although enhanced N deposition often stimulates forest growth and carbon sequestration (Högberg, 2007; De Vries et al., 2009; Thomas et al., 2010), the expected growth acceleration can be diminished in P-limited forest ecosystems (Braun et al., 2010; Crowley et al., 2012; Li et al., 2016). Phosphorus limitation may not only constrain future forest growth in response to N deposition but also lower the projected $CO_2$ fertilization effects on primary productivity (Wieder et al., 2015). Modelling results have also indicated that terrestrial carbon sequestration in China showed a lower response to per unit N deposition in recent years (Tian et al., 2011). Unless efficient measures are taken to reduce anthropogenic N emissions in China, the threats of human-induced nutrient imbalance to the health and function of forest ecosystems may keep increasing in the next decades.

## 5. Conclusions

Our results show relatively high P inputs by bulk precipitation and throughfall, but they are accompanied by even much higher N inputs by bulk precipitation and throughfall, respectively. High N:P ratios in bulk deposition and throughfall suggest an imbalance of P and N deposition, which is likely resulting in a shift towards P limitation in forests especially in southern China. Moreover, spatial patterns of P and N deposition both showed a strong power-law increase with closer distance to the nearest large cities, implying a significant alternation of regional P and N cycling by rapid urbanization in China.

Although China has started to cut down $NO_x$ emissions since early 2010s (Twelfth Five-year Plan, A full version of the plan is available at http://news.xinhuanet.com/politics/2011-03/16/c_121193916.htm), the absence of $NH_3$ regulation policy and an increase in meat and dairy consumption may further enhance emissions and deposition of reduced N in the future. High levels of N deposition are expected to continue in the next decades and may further enlarge the nutrient imbalance in China's forest ecosystems. In order to gain a better understanding of the sources, composition and rates of P deposition as



well as its ecological effects, monitoring networks of atmospheric deposition (e.g. Nationwide Nitrogen Deposition Monitoring Network, NNDMN, Xu et al., 2015; Chinese Ecosystem Research Network, CERN, Zhu et al., 2015) are encouraged to include measurements of P deposition across the country based on standardized sampling protocols and analytical methods. Field observations and manipulated experiments should be conducted to assess the impacts of nutrient imbalance on the health and function of China's forest ecosystems. Moreover, better forest management strategies should be adopted to avoid loss of forestry production from the human-induced P and N imbalance.

**Author contribution**

E. Du. and W. de Vries conceived the idea. E. Du. conducted data analysis and prepared the manuscript. W. de Vries, Y. Jiang, W. Han, X. Liu and Z. Yan reviewed and edited the manuscript.

**Acknowledgement**

This study was supported by National Natural Science Foundation of China (Nos. 31400381 and 40425007), Youth Scholars Program of Beijing Normal University (No. 2015NT08) and Open Foundation of Key Laboratory for Earth Surface Processes of the Ministry of Education (Peking University, No. 201401).

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





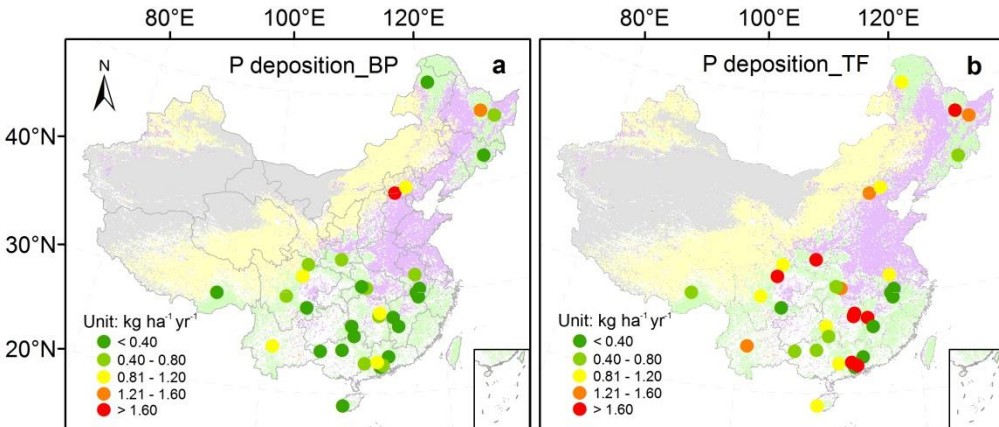

Figure 1. Spatial patterns of P deposition (kg ha$^{-1}$ yr$^{-1}$) in a) bulk precipitation (BP) and b) throughfall (TF) in China's forests. The background shadows in light green, light yellow, light purple and light grey indicate the distributions of forest, grassland (semiarid region), cropland and non-vegetated land, respectively.



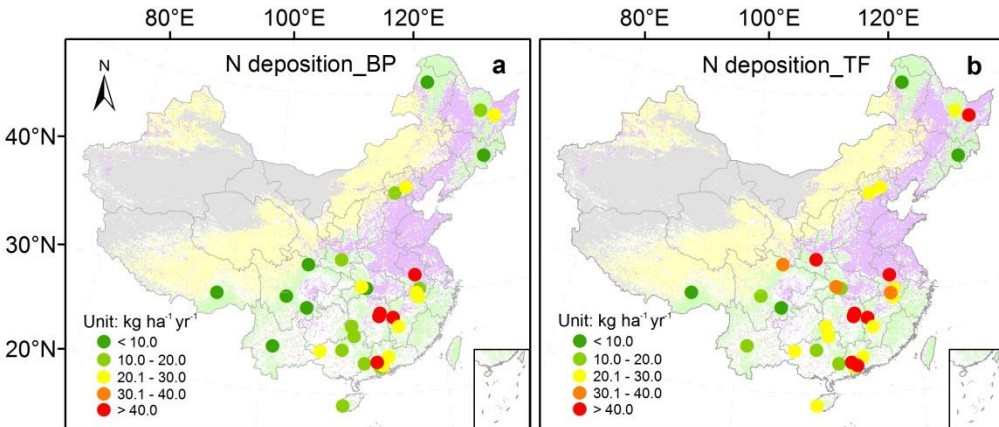

Figure 2. Spatial patterns of N deposition (kg ha$^{-1}$ yr$^{-1}$) in a) bulk precipitation (BP) and b) throughfall (TF) in China's
forests. The background shadows in light green, light yellow, light purple and light grey indicate the distributions of forest,
grassland (semiarid region), cropland and non-vegetated land, respectively.



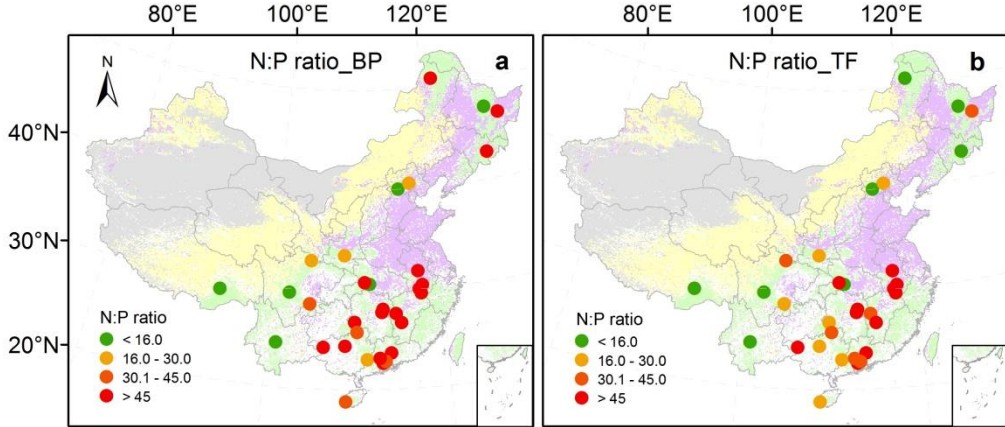

Figure 3. Spatial patterns of N:P ratios in a) bulk precipitation (BP) and b) throughfall (TF) in China's forests. The background shadows in light green, light yellow, light purple and light grey indicate the distributions of forest, grassland (semiarid region), cropland and non-vegetated land, respectively.





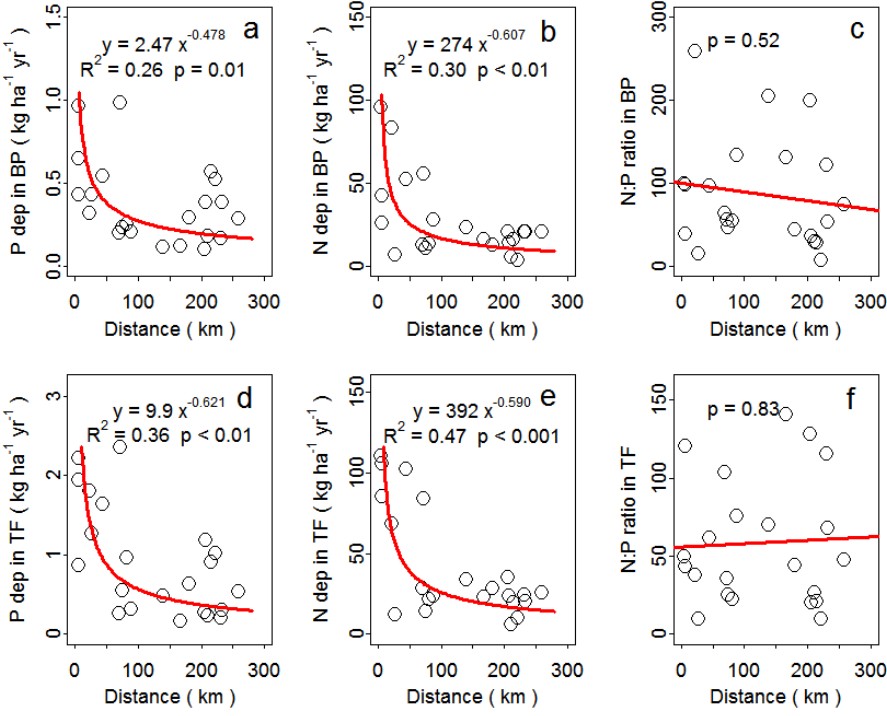

Figure 4. Changes in P and N deposition (kg ha$^{-1}$ yr$^{-1}$) and N:P ratios in bulk precipitation (BP) and throughfall (TF) with the distance to the nearest large cities based on datasets of sites far from semiarid regions in China.





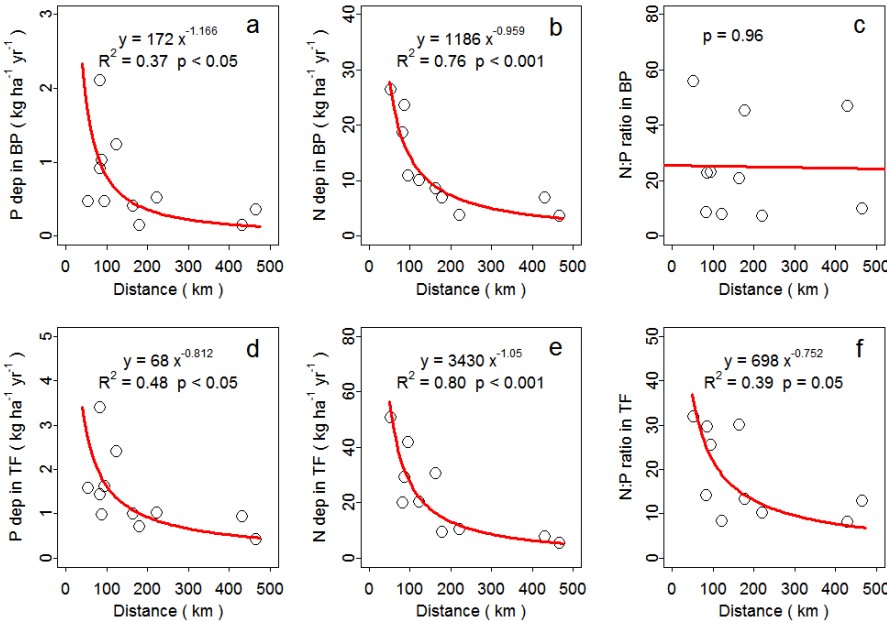

Figure 5. Changes in P and N deposition (kg ha$^{-1}$ yr$^{-1}$) and N:P ratios in bulk precipitation (BP) and throughfall (TF) with the distance to the nearest large cities based on datasets of sites nearby semiarid regions in China.