# Peer review of "Imbalanced phosphorus and nitrogen deposition in China's forests"

_Atmospheric Chemistry and Physics, 2015_

## Referee Comment (RC1) · Anonymous Referee #1 · 22 Mar 2016

This is a potentially publishable article about an interesting topic (N:P ratios in deposition in China), but there are a few points that need modification and/or justification before the paper can be accepted.

Major point: How can you deduce dry deposition from throughfall and precipitation concentrations? This needs to be better explained or justified, and the following points are where I get confused.

"Phosphorus and N fluxes in the bulk precipitation and throughfall were estimated according to the volume-weighted mean concentration and annual precipitation." So you are taking your measured concentrations in each precip and throughfall and multiplying by annual precip? But not all the precip comes out in throughfall does it?

Canopy captured dry deposition was estimated as the difference between throughfall

deposition and bulk deposition." How can you get dry deposition from the difference in two wet deposition numbers?

Later on you try to use this: "Based on the P enrichment in throughfall versus bulk precipitation, the mean canopy captured dry P deposition was estimated as high as 0.46 kg P ha-1 yr-1 in China's forests, being significantly higher than bulk deposition (0.38 kg P ha-1 yr-1) (paired t-test, df = 32, p < 0.01)." So I think you need to be a bit clearer: maybe make a skematic of your assumptions?

Again you say something here that doesn't make sense to me: "Although throughfall may still underestimate total deposition due to canopy uptake (Draaijers et al., 1996; Sparks, 2009), it is a better proxy of total N deposition than bulk deposition because bulk deposition only accounts for 63% of throughfall deposition."

Are you only concerned with soluble P?

It is important to clarify these points. If you are going to argue that from two wet deposition numbers you can deduce dry deposition and total deposition, then maybe you want to verify at a point where you have all three numbers to show that your approach works?

Your paper could be publishable with only wet deposition numbers, but I think you just need to be clear about what your results are.

Secondary points: In the comparison to the model results there are two issues:

i. "Anthropogenic sources have been traditionally thought to make only a small contribution to P deposition (Okin et al., 2004; Mahowald et al., 2008). In contrast, our results emphasize that anthropogenic sources near large cities can have significant impacts on the spatial patterns of regional P deposition. The urban hotpot of P deposition might be derived from intensive combustion-related emissions near urban areas (Wang et al., 2015) and a short-distance transfer of P-containing aerosols from P-rich farmland soils (Anderson et al., 2006)." The problem with this conclusion is that it contradicts the observations here that the highest P deposition comes from regions close to the arid sources of dust: so your gradient suggests the dominant sources are dust related. Even using the smaller anthropogenic source of Mahowald et al., 2008, anthropogenic sources should dominate near cities in China (Figure 5h: Mahowald et al., 2008). So I would actually argue that your data supports Mahowald et al., 2008's budgets quite well. In contrast, from Wang et al., 2015, you shouldn't be seeing a P deposition gradient across China (Figure 3) from the arid regions, but only the combustion hot spots.

You could actually be a bit quantitative and do the comparison against the model output from these two models, to discriminate between them? See if there is a statistically significant difference in the comparison?

ii. Both the model of Mahowald et al., 2008 and Wang et al., 2015 only include long range transported P, so less than 10um or 20um particles. Your results, in contrast, should be including all P (unless you are only looking at soluble P, which should then not be compared to Wang et al., 2015 and only the soluble P in Mahowald et al., 2008). In forests, most of the P is in much larger mode particles, locally generated, according to previous observational synthesis (and the observational studies within them; e.g. Tipping et al., 2014). So it is likely you have a mismatch in the P deposition you are comparing (see Mahowald et al., 2008; Neff et 2009; Brahney et al., 2015 for more discussion on how to compare observational P fluxes across sizes to models or not). Notice that Wang et al., 2015 is very careful about the size of their emitted combustion P particles, but not at all careful about the size of the particles in the deposition P dataset they use (Tipping et al., 2014; which is dominated by large local generated primary biogenic particles, according to that study), compared to the particles that they are modeling (long range transported particles <20um). They actually use the mismatch in particle size and the resulting mismatch in deposition magnitude to deduce the extra-large combustion source they postulate, which makes that study problematic in its conclusions.

Minor points:

Section 3.1: I am not sure what I am supposed to get out of knowing that the data follows a log-normal distribution for P or N or N:P. Isn't this what we expect? I think this would be better illustrated with a figure of the distribution, if you think it's important enough to discuss?

"Spatial patterns of P deposition via bulk precipitation and throughfall showed large heterogeneity" This is a factor of 4. Globally, the deposition varies by several orders of magnitude, so I'm not sure I would call this a large spatial hetereogeneity. Maybe just say the values vary by a factor of 4.

Some English errors, especially in the few last lines of the introduction, should be fixed.

---

## Referee Comment (RC2) · Anonymous Referee #2 · 24 Mar 2016

Review:

This is an interesting paper, discussing a previously not available dataset on phosphorus and nitrogen deposition fluxes on forests (through fall and bulk deposition). The paper is well written, and the context and relevance are well explained. The issue of increasing dis-balance of phosphorous to nitrogen ratios may further worsen, if fire from coal and biofuel use in China will be reduced, and/or particulate emissions will be better controlled. This issues could warrant some further discussion.

I see some major weaknesses of this study:

1) The assumption that bulk minus through equals dry deposition is rather challenging. Throughfall measurements are easily compromised by input from the canopy, it is not clear what was done to prevent this. For N bulk deposition may not capture input

potential significant input from gaseous nitric acid- and may in this sense not be a good proxy for total deposition. Some discussion on the specific situation in China is warranted. I guess the applicability of the approach to estimate phosporus depostion is even less well known. At any rate a better quantification of errors is needed.

2) the constraint of only using bulk deposition and throughfall observations (where both are available), is providing a very limited amount of observations, and raises questions about representativity for larger regions. I understand that a wider dataset of deposition measurements are available in other ecosystems (e.g. the authors mention that N-dep is relatively well known), and I would recommend to analyse also these in the context of this core set of depositions over forests. To what extent are the forest observation consistent with nearby depositions over other regions? Are the same urban-rural decay of depositions observable also in other datasets? What would wet-versus bulk versus throughfall tell?

3) As the motivation of the study is to point to a dis-balance in P:N ratios- I wonder how the numbers in this study compare to published and modeled deposition maps of N and P, used as input to vegetation models. For instance p6/l3 mentions a number of 4.6 Tg N/yr deposition on Chinese forests. In my impression this is not very different from model estimates, but this can be corroborated. How would the estimate of P deposition compare to current estimates (e.g. Wang 2015; Mahowald; 2008).

Nevertheless, altogether an important dataset and analysis, which I would recommend to publish in ACP, after duly accounting for these comments.

Minor comment:

P 3/l 9: what is taken city-centre or city-boundary? How defined?

p. 6 l 24 were is this number coming from?

---

## Author Comment (AC1) · 23 Apr 2016

Enzai Du[1], Wim de Vries[2, 3], Wenxuan Han[4], Xuejun Liu[4], Zhengbing Yan[5], Yuan Jiang[1]

[1]State Key Laboratory of Earth Surface Processes and Resource Ecology, and College of Resources Science & Technology, Beijing Normal University, Xinjiekouwai Street 19#, Beijing, 100875, China

[2]Environmental Systems Analysis Group, Wageningen University, PO Box 47, 6700 AA Wageningen, The Netherlands

[3]Alterra, Wageningen University and Research Center, PO Box 47, 6700 AA Wageningen, the Netherlands

[4]College of Resources and Environmental Sciences, China Agricultural University, Beijing, 100193, China

[5]Department of Ecology, and Key Laboratory for Earth Surface Processes of the Ministry of Education, Peking University, Beijing, 100871, China

*Correspondence to*: Enzai Du (enzaidu@bnu.edu.cn) and Yuan Jiang (jiangy@bnu.edu.cn)

This is a potentially publishable article about an interesting topic (N:P ratios in deposition in China), but there are a few points that need modification and/or justification before the paper can be accepted.

*Reply: Thank you. We have revised the manuscript according to your suggestions.*

Major point: How can you deduce dry deposition from throughfall and precipitation concentrations? This needs to be better explained or justified, and the following points are where I get confused.

*Reply: Before going into the detailed questions we like to give a general overview of our assumptions, which we have now also included in the manuscript. Canopy-captured dry deposition accumulates during periods without precipitation and it is washed down during precipitation events. Here we calculated canopy-captured dry deposition as the difference between bulk deposition and the estimated total deposition. In our original manuscript, total deposition was estimated based on throughfall concentrations and total (bulk) precipitation but in the revised manuscript we now multiply it with the sum of throughfall and stemflow water fluxes. To estimated the sum water flux of throughfall and stemflow, we establish a new database of observed throughfall and stemflow (including data from 28 forested stands from 26 sites, see Figure R1 as below) and explored an empirical equation between precipitation and the sum of throughfall and stemflow (TS =0.88×precipitation-64.6, $R^2$=0.97). The sum of*

*throughfall and stemflow water fluxes for each forested site was then estimated from the available bulk precipitation data based on this empirical equation. We realize that our estimate of canopy-captured dry deposition is lower than total dry deposition because a proportion of dry deposition is already included in bulk deposition and our estimate of total deposition is likely underestimated due to the neglect of canopy uptake (Reddy and Majmudar, 1983; Draaijers et al., 1996; Sparks, 2009) and stemflow enrichment (N & P concentrations in stemflow are generally higher than that in throughfall). We now mention this explicitly in the discussion and we will include detailed information on the method and uncertainty in the method section in the revised manuscript.*

[Figure]

*Figure R1. Distribution of 26 sites with measured data of throughfall and stemflow.*

"Phosphorus and N fluxes in the bulk precipitation and throughfall were estimated according to the volume-weighted mean concentration and annual precipitation." So you are taking your measured concentrations in each precip and throughfall and multiplying by annual precip? But not all the precip comes out in throughfall does it?

*Reply: This is a correct comment. Indeed not all precipitation comes out in throughfall and we now do not use total (bulk) precipitation any more but the sum of throughfall and stemflow water fluxes estimate (see above). On an annual basis, total deposition can be estimated as the sum of N/P fluxes in throughfall and stemflow as well as canopy exchange (see Figure R2 as below). Due to a lack of measured data on canopy exchange and stemflow*

*concentrations, we approximately estimated total deposition by multiplying the volume-weighted mean N/P concentration in throughfall with the sum of throughfall and stemflow water fluxes in the revised manuscript. Our results generally underestimate total deposition because concentrations in throughfall usually are much lower than those in stemflow and tree foliage can take up tree foliage can take up a small proportion of soluble N and P in rainwater and gaseous N (part of dry deposition) (Reddy and Majmudar, 1983; Draaijers et al., 1996; Sparks, 2009). More detailed information on the method and its uncertainty will be included in the revised manuscript.*

[Figure]

*Figure R2. Partition of nitrogen or phosphrus flows when rainwater passes through forest canopy.*

Canopy captured dry deposition was estimated as the difference between throughfall deposition and bulk deposition." How can you get dry deposition from the difference in two wet deposition numbers? Later on you try to use this: "Based on the P enrichment in throughfall versus bulk precipitation, the mean canopy captured dry P deposition was estimated as high as 0.46 kg P ha-1 yr-1 in China's forests, being significantly higher than bulk deposition (0.38 kg P ha-1 yr-1) (paired t-test, df = 32, p < 0.01)." So I think you need to be a bit clearer: maybe make a skematic of your assumptions?

*Reply: Canopy-captured dry deposition is defined as the amounts of dry deposition captured and accumulated in forest canopy during the period without precipitation events. Therefore, we calculated the canopy-captured dry deposition as the difference between bulk deposition and the estimated total deposition (as described above). The new result indicate that mean canopy captured dry P deposition was estimated to be 0.40 kg P ha$^{-1}$ yr$^{-1}$ in China's*

*forests, being comparable to bulk deposition (0.38 kg P ha$^{-1}$ yr$^{-1}$) (paired t-test, df = 32, p = 0.36). We will include more detailed information on the method in the method section and discussed uncertainties of our results in the revised manuscript.*

Again you say something here that doesn't make sense to me: "Although throughfall may still underestimate total deposition due to canopy uptake (Draaijers et al., 1996; Sparks, 2009), it is a better proxy of total N deposition than bulk deposition because bulk deposition only accounts for 63% of throughfall deposition."

*Reply: The last part of this sentence may have caused confusion but our point is that even though throughfall may underestimate total deposition it is a better proxy of total N deposition than bulk deposition that makes a much larger underestimation. We now changed the sentence to: "Although our estimate based on annual sum of throughfall and stemflow water fluxes and total N concentrations in throughfall is still lower than the factual total deposition because tree foliage can take up a small proportion of gaseous N (part of dry deposition) and soluble N in rainwater (Draaijers et al., 1996; Sparks, 2009) and the stemflow N flux is likely underestimated, it is a better proxy of total N deposition than bulk deposition." In addition, the uncertainties of our estimates are also discussed in the revised Discussion section.*

Are you only concerned with soluble P?

*Reply: Our analysis focused on total P concentrations, which were measured after digestion and include dissolved inorganic forms, dissolved organic forms and particulate forms. This is mentioned in the text of the method section.*

It is important to clarify these points. If you are going to argue that from two wet deposition numbers you can deduce dry deposition and total deposition, then maybe you want to verify at a point where you have all three numbers to show that your approach works? Your paper could be publishable with only wet deposition numbers, but I think you just need to be clear about what your results are.

*Reply: Thank you for your suggestions. More detailed information on the method and uncertainty has been included in the revised manuscript (also see replies above).*

Secondary points: In the comparison to the model results there are two issues:

i. "Anthropogenic sources have been traditionally thought to make only a small contribution to P deposition (Okin et al., 2004; Mahowald et al., 2008). In contrast, our results emphasize that anthropogenic sources near large cities

can have significant impacts on the spatial patterns of regional P deposition. The urban hotpot of P deposition might be derived from intensive combustion-related emissions near urban areas (Wang et al., 2015) and a short-distance transfer of P-containing aerosols from P-rich farmland soils (Anderson et al., 2006)." The problem with this conclusion is that it contradicts the observations here that the highest P deposition comes from regions close to the arid sources of dust: so your gradient suggests the dominant sources are dust related. Even using the smaller anthropogenic source of Mahowald et al., 2008, anthropogenic sources should dominate near cities in China (Figure 5h: Mahowald et al., 2008). So I would actually argue that your data supports Mahowald et al., 2008's budgets quite well. In contrast, from Wang et al., 2015, you shouldn't be seeing a P deposition gradient across China (Figure 3) from the arid regions, but only the combustion hot spots. You could actually be a bit quantitative and do the comparison against the model output from these two models, to discriminate between them? See if there is a statistically significant difference in the comparison?

*Reply: Our results show significantly higher P deposition nearby than far from semiarid regions and this indeed supports Mahowald et al., 2008's conclusions that large-scale gradients of P deposition are dust dominated. At regional scale, either far from or nearby the semiarid regions, our results indicate that P deposition showed a significant power-law increase with closer distance to the nearest large cities and this supports Wang et al., 2015's conclusion that anthropogenic contribution to P deposition can be substantial. Overall, our results don't contradict the conclusions either by Mahowald et al.(2008) or Wang et al. (2015). Moreover, the modelling results usually have a very coarse resolution and are not suitable for quantitative comparison with our datasets. We have revised the text to make this point more clear.*

ii. Both the model of Mahowald et al., 2008 and Wang et al., 2015 only include long range transported P, so less than 10um or 20um particles. Your results, in contrast, should be including all P (unless you are only looking at soluble P, which should then not be compared to Wang et al., 2015 and only the soluble P in Mahowald et al., 2008). In forests, most of the P is in much larger mode particles, locally generated, according to previous observational synthesis (and the observational studies within them; e.g. Tipping et al., 2014). So it is likely you have a mismatch in the P deposition you are comparing (see Mahowald et al., 2008; Neff et 2009; Brahney et al., 2015 for more discussion on how to compare observational P fluxes across sizes to models or not). Notice that Wang et al., 2015 is very careful about the size of their emitted combustion P particles, but not at all careful about the size of the particles in the deposition P dataset they use (Tipping et al., 2014; which is dominated by large local generated primary biogenic particles, according to that study), compared to the particles that they are modeling (long range

transported particles <20um). They actually use the mismatch in particle size and the resulting mismatch in deposition magnitude to deduce the extra-large combustion source they postulate, which makes that study problematic in its conclusions.

*Reply: In the discussion section, we mostly compared our results with other assessments based on observations (e.g. Tipping et al., 2014, Vet et al., 2014). We have already recognized that different studies may focus on different components of P and therefore we are very careful when making comparison with other results.*

Minor points:

Section 3.1: I am not sure what I am supposed to get out of knowing that the data follows a log-normal distribution for P or N or N:P. Isn't this what we expect? I think this would be better illustrated with a figure of the distribution, if you think it's important enough to discuss?

*Reply: The description of datasets (e.g. arithmetic mean for a normal distribution and geometric mean for a log-normal distribution) should be according to their distribution. Therefore, we tested the distribution of the datasets at the beginning.*

"Spatial patterns of P deposition via bulk precipitation and throughfall showed large heterogeneity" This is a factor of 4. Globally, the deposition varies by several orders of magnitude, so I'm not sure I would call this a large spatial hetereogeneity. Maybe just say the values vary by a factor of 4.

*Reply: We have done so.*

Some English errors, especially in the few last lines of the introduction, should be fixed.

*Reply: We have checked and corrected all the linguistics errors in the manuscript.*

*References:*

*Draaijers, G. P. J., Erisman, J. W., Sprangert, T. and Wyers, G. P. 1996. The application of throughfall measurements for atmospheric deposition monitoring. Atmospheric Environment: 30(19), 3349–3361.*

*Reddy, S. E., Majmudar, A. M. 1983. Response of mango (Mangifera indica L.) to foliar application of phosphorus. Fertilizer Research 4(3): 281-285.*

*Sparks, J. P.2009. Ecological ramifications of the direct foliar uptake of nitrogen. Oecologia: 159, 1–13.*

---

## Author Comment (AC2) · 23 Apr 2016

Enzai Du1, Wim de Vries2, 3, Wenxuan Han4, Xuejun Liu4, Zhengbing Yan5, Yuan Jiang1

1State Key Laboratory of Earth Surface Processes and Resource Ecology, and College of Resources Science & Technology, Beijing Normal University, Xinjiekouwai Street 19#, Beijing, 100875, China
2Environmental Systems Analysis Group, Wageningen University, PO Box 47, 6700 AA Wageningen, The Netherlands
3Alterra, Wageningen University and Research Center, PO Box 47, 6700 AA Wageningen, the Netherlands
4College of Resources and Environmental Sciences, China Agricultural University, Beijing, 100193, China
5Department of Ecology, and Key Laboratory for Earth Surface Processes of the Ministry of Education, Peking University, Beijing, 100871, China

Correspondence to: Enzai Du (enzaidu@bnu.edu.cn) and Yuan Jiang (jiangy@bnu.edu.cn)

**Review:**

This is an interesting paper, discussing a previously not available dataset on phosphorus and nitrogen deposition fluxes on forests (through fall and bulk deposition). The paper is well written, and the context and relevance are well explained. The issue of increasing dis-balance of phosphorous to nitrogen ratios may further worsen, if fire from coal and biofuel use in China will be reduced, and/or particulate emissions will be better controlled. This issues could warrant some further discussion.

*Reply: Thank you very much. In the revised manuscript, we will include a short discussion on future changes in the imbalance of P and N deposition in the ending paragraph.*

I see some major weaknesses of this study:

1) The assumption that bulk minus through equals dry deposition is rather challenging. Throughfall measurements are easily compromised by input from the canopy, it is not clear what was done to prevent this. For N bulk deposition may not capture input potential significant input from gaseous nitric acid- and may in this sense not be a good proxy for total deposition. Some discussion on the specific situation in China is warranted. I guess the applicability of the approach to estimate phosphorus deposition is even less well known. At any rate a better quantification of errors is needed.

Reply: Thanks for this useful and correct comment. Canopy-captured dry deposition accumulates during periods without precipitation and it is washed down during precipitation events. Therefore, we calculated canopy-captured dry

deposition as the difference between bulk deposition and the estimated total deposition. On an annual basis, total deposition can be estimated as the sum of N/P fluxes in throughfall and stemflow as well as canopy exchange. In the revised manuscript, we approximate total deposition by multiplying the volume-weighted mean N/P concentration in throughfall with the sum of throughfall and stemflow water fluxes because of a lack of measured data on canopy exchange and stemflow concentrations. We realize that our results generally underestimate total deposition because concentrations in throughfall usually are lower than those in stemflow (underestimate of N/P fluxes in stemflow) and tree foliage can take up tree foliage can take up a small proportion of soluble N and P in rainwater and gaseous N (part of dry deposition) (Reddy and Majmudar, 1983; Draaijers et al., 1996; Sparks, 2009). We also realize that our estimate of canopy-captured dry deposition is lower than total dry deposition because a proportion of dry deposition is already included in bulk deposition and the total deposition is likely underestimated due to the neglect of canopy uptake (Reddy and Majmudar, 1983; Draaijers et al., 1996; Sparks, 2009) and the underestimate of stemflow N/P fluxes. More detailed information on the method and uncertainty will therefore be included in the revised manuscript to answer your concerns.

2) the constraint of only using bulk deposition and throughfall observations (where both are available), is providing a very limited amount of observations, and raises questions about representativity for larger regions. I understand that a wider dataset of deposition measurements are available in other ecosystems (e.g. the authors mention that N-dep is relatively well known), and I would recommend to analyse also these in the context of this core set of depositions over forests. To what extent are the forest observation consistent with nearby depositions over other regions? Are the same urban-rural decay of depositions observable also in other datasets? What would wet-versus bulk versus throughfall tell?

Reply: Thank you very much for your suggestions. The database used in this study has included most of the forest sites where nutrient deposition has been observed. Moreover, these sites are evenly distributed in the forested areas in China (Figure S1). We believe that the datasets used in this study currently is the best to represent the observations of N and P deposition in China's forest. As indicated by the title, our analysis has been focused on China's forest ecosystems where the response of C sequestration to atmospheric nutrient deposition currently is a core topic. In addition, our estimates of total deposition are based on throughfall N/P concentrations and annual precipitation and this method is not applicable in other types of ecosystem. Nevertheless, a previous analysis on wet N deposition across China has also indicated an urban-rural decay of N deposition (Du, E.Z. and Liu, X.J.: High rates of wet nitrogen deposition in China: A synthesis. In: Sutton, M. A., Mason, K. E., Sheppard, L. J., Sverdrup, H., Haeuber, R., Hicks W. K. (eds.) Nitrogen Deposition, Critical Loads and Biodiversity. Springer, Netherlands, pp 49–56, 2014.). 3) As the motivation of the study is to point to a dis-balance in P:N ratios- I wonder how the numbers in this study compare to published and modeled deposition maps of N and P, used as input to vegetation models. For instance p6/13 mentions a number of 4.6 Tg N/yr deposition on Chinese forests. In my impression this is not very different from model estimates, but this can be corroborated. How would the estimate of P deposition compare to current estimates (e.g. Wang 2015; Mahowald; 2008). Nevertheless, altogether an important dataset and analysis, which I would recommend to publish in ACP, after duly accounting for these comments.

Reply: There have been rare literatures which focus on total N and P inputs into China's forests. This hinders the comparison between our estimates with others. Nevertheless, our results in China's forests are comparable to the ranges of modelled P deposition in China (Wang et al., 2015). This is now indicated in the text. Unfortunately, the modelling results by Mahowald et al (2008) have a very coarse resolution and are not suitable for comparison with our datasets.

**Minor comment:**

P 3/1 9: what is taken city-centre or city-boundary? How defined?

Reply: The distance between the sampling site and the centre of the nearest large city (non-agricultural population > 0.5 million) was derived using Google Earth for Microsoft Windows. The city centre is defined as the geometric centre of the city area. We have included the information in the revised manuscript.

p. 6124 were is this number coming from?

Reply: The number 63% is the percentage that bulk N deposition (16.5 kg N ha-1 yr-1) accounts for the total N deposition (26.2 kg N ha-1 yr-1) estimated based on annual precipitation and total N concentrations in throughfall. In the revised manuscript, we re-estimated total deposition by multiplying the volume-weighted mean N/P concentration in throughfall with the sum water flux of throughfall and stemflow (see replies above). The updated results indicate that bulk N deposition (16.5 kg N ha-1 yr-1) accounts for 76% of the total N deposition (21.5 kg N ha-1 yr-1).

---

## Author Response (AR2)

**Reply to comments on "Imbalanced phosphorus and nitrogen deposition in China's forests" by Anonymous Referee #1**

Enzai Du[1], Wim de Vries[2, 3], Wenxuan Han[4], Xuejun Liu[4], Zhengbing Yan[5], Yuan Jiang[1]

[1]State Key Laboratory of Earth Surface Processes and Resource Ecology, and College of Resources Science & Technology, Beijing Normal University, Xinjiekouwai Street 19#, Beijing, 100875, China

[2]Environmental Systems Analysis Group, Wageningen University, PO Box 47, 6700 AA Wageningen, The Netherlands

[3]Alterra, Wageningen University and Research Center, PO Box 47, 6700 AA Wageningen, the Netherlands

[4]College of Resources and Environmental Sciences, China Agricultural University, Beijing, 100193, China

[5]Department of Ecology, and Key Laboratory for Earth Surface Processes of the Ministry of Education, Peking University, Beijing, 100871, China

*Correspondence to*: Enzai Du (enzaidu@bnu.edu.cn) and Yuan Jiang (jiangy@bnu.edu.cn)

Review

This is still a potentially publishable manuscript, however there are a few edits that need to be made.

*Reply: Thank you. We have revised the manuscript according to your suggestions.*

I would like to heavily encourage the authors to submit a revised manuscript with all the changes highlighted and indicate in the response to reviewers exactly the text changed. Because this was not done, I had to reread the entire manuscript, and thus, perhaps I have found elements that were in the first version, but would have missed if the authors has highlighted their changes more effectively.

*Reply: We are sorry that we didn't submit a revised manuscript with highlighted changes last time. This time we include a marked-up manuscript version showing the changes we have made.*

The question of what kind of deposition the authors are looking at is better explained in the methods section: I see there the highlighted changes. However, I am still not sure that these aren't the soluble N and P, instead of total deposition. Because of the methods shown, and the uncertainty about what exactly this deposition is, I would like to encourage the authors to be more clear that they are using wet deposition in both the abstract, introduction and

conclusions.

*Reply: We believe that we describe in the methods section how we approximated total P and N deposition. As shown in the first part of the method section, our analysis is based on datasets of total P and total N contents in bulk precipitation and throughfall, which are assessed after digestion and include dissolved inorganic forms, dissolved organic forms and particulate forms. We then mention (see page 3 line 23-26) that on an annual basis, total deposition equals the sum of N or P fluxes in throughfall, stemflow and canopy exchange but that we approximate total deposition by multiplying the volume-weighted mean concentration in throughfall with the sum water fluxes of throughfall and stemflow. We then mention (see top of page 4) that this approach causes a slight underestimate of total deposition since tree foliage can take up a small proportion of soluble N and P and because throughfall concentrations of N or P are generally were lower than those in stemflow. We thus believe that the method section clearly shows how we used total (not soluble) N and P concentrations in bulk precipitation and throughfall to estimated bulk deposition, total deposition and canopy-captured dry deposition.*

In the response to the reviewers, the authors indicate that they are very interested in ecological impacts in managed forests, which is really interesting and important and should be added to the introduction, and more so to the results and conclusions. If they are interested in ecological impacts, aren't they more interested in N:P ratios in the deposition? Following the ideas of Brahney et al., 2016, shouldn't they look at how N:P ratios in vegetation relate to the deposition ? The P deposition is much more effective (14x), because less P is required. How does that change the conclusions of this study?

*Reply: Thanks for the suggestions. The introduction section has included some information on the possible effects of imbalanced P and N deposition. We now added information on the paired analysis by Brahney et al. (2015) on N:P ratios in remote alpine lakes with N:P ratios in atmospheric deposition. We mention however, that such a relationship can expected in remote lakes where atmospheric deposition is the dominant nutrient source, whereas in forest ecosystems atmospheric P and N deposition is only a relatively small part of the total nutrient supply, most of which is derived from mineralization and/or weathering. A lack of corresponding datasets of N:P ratios in forest vegetation hinders us to test this correlation, but we expect a less clear relationships. Nevertheless, based on an analysis of literature, we expect that "the role of P deposition is relevant because high-level N deposition, initially stimulating forest growth, is likely shifting large areas of China's forests towards P limitation". In the discussion section, further analysis has been made on the imbalance of P and N deposition. The N:P ratios in bulk precipitation and throughfall is more than twice of that in tree leaves (geometric mean N:P ratio = 15) (Han et al.,*

*2005) and three times of that in current-year twigs (geometric mean N:P ratio = 10.6) (Yao et al., 2015). It is also much higher than critical N:P ratios related to relative P limitation in view of forest growth, which are near 15 for coniferous forests and near 25 for deciduous forests (after Mellert and Gätlein, 2012).*

"Based on measurements of P content in various fuels and estimates of the P partitioning during combustion processes, a recent assessment indicates that anthropogenic sources, such as combustion-related emissions, can contribute to over 50% of the global atmospheric P and make a substantial contribution to global P deposition (Wang et al., 2015)." This is not correct. They get this large number by ignoring size differentiation between their model (<20um) and locally produced primary biogenic aerosols, and assuming that deposition in forests generated by locally produced biogenic aerosols (in Tipping et al., 2013) comes from combustion instead. Using different assumptions, Brahney et al., 2015 shows that one can match the Tipping et al., 2013 data, as well as other data constraining the size of the combustion sources (as done in Mahowald et al., 2008, but not done in Wang et al., 2015). Please rewrite to "Based on assumptions of the P partitioning during combustion processes and local deposition, a recent assessment hypothesizes that anthropogenic sources, such as combustion-related emissions, can contribute to over 50% of the global atmospheric P budget (Wang et al., 2015), although this result is very sensitive to assumptions about size distribution (Brahney et al., 2016)."

*Reply: Thank you for this correction. We have done according to the suggestion.*

"Anthropogenic sources have been traditionally thought to make only a small contribution to P deposition (Okin et al., 2004; Mahowald et al., 2008). In contrast, our results emphasize that anthropogenic sources near large cities can have significant impacts on the spatial patterns of regional P deposition. The urban hotpot of P deposition might be derived from intensive combustion-related emissions near urban areas (Wang et al., 2015) and a short-distance transfer of P-containing aerosols from P-rich farmland soils (Anderson et al., 2006)." Again, as in the first version, the Mahowald et al., 2008 study also has hotspots close to cities, so it is not correct that you need huge P sources from combustion to get this result. Also, you get maximum values close to the arid regions. Please be more clear on this. I would suggest something like: "Anthropogenic sources have been traditionally thought to make only a small contribution to P deposition (Okin et al., 2004; Mahowald et al., 2008), and yet can still result in local hotspots of deposition (e.g. Mahowald et la., 2008: Wang et al., 2015) along with P from arid regions, which is consistent with our study. The urban hotpot of P deposition might be derived from intensive combustion-related emissions near urban areas (Wang et al., 2015) and a short-distance transfer of P-containing aerosols from P-rich farmland soils

(Anderson et al., 2006)."

*Reply: Thank you. We have revised the text almost completely as suggested, i.e. "
[revised manuscript text omitted]

---

## Author Response (AR3)

**Reply to comments on "Imbalanced phosphorus and nitrogen deposition in China's forests" by the Editor**

Enzai Du[1], Wim de Vries[2, 3], Wenxuan Han[4], Xuejun Liu[4], Zhengbing Yan[5], Yuan Jiang[1]

[1]State Key Laboratory of Earth Surface Processes and Resource Ecology, and College of Resources Science & Technology, Beijing Normal University, Xinjiekouwai Street 19#, Beijing, 100875, China

[2]Environmental Systems Analysis Group, Wageningen University, PO Box 47, 6700 AA Wageningen, The Netherlands

[3]Alterra, Wageningen University and Research Center, PO Box 47, 6700 AA Wageningen, the Netherlands

[4]College of Resources and Environmental Sciences, China Agricultural University, Beijing, 100193, China

[5]Department of Ecology, and Key Laboratory for Earth Surface Processes of the Ministry of Education, Peking University, Beijing, 100871, China

*Correspondence to*: Enzai Du (enzaidu@bnu.edu.cn) and Yuan Jiang (jiangy@bnu.edu.cn)

Comments to the Author:

Dear authors- you have to a large extent sufficiently responded to the request of reviewer 1. Nevertheless, I suggest that at the beginning of section 2.1 you start with presenting the definitions of total, bulk, wet and dry deposition, and whether or not it includes insoluble N and P. I understand wet deposition per se has not been used in this analysis, but there is a community very used to see these (more strictly defined) measurements. This will avoid possible confusion with readers as to what has been assumed. thank you

*Reply: Thank you for the suggestions.*

*At the beginning of section 2, we now added the definitions of total, wet, dry, and bulk deposition by adding the following paragraph "In terms in physics, total atmospheric deposition of elements is distinguished in (i) wet deposition, being their deposition in rain and snow, and (ii) dry deposition, being their deposition in gases, aerosols and particles. However, most assessments in China have been based on measurements of bulk deposition, which refers to precipitation samples collected by continuously opening funnels and includes wet deposition and part of dry deposition because some dry-deposited compounds are inevitably collected during intervening dry periods. In this study, we synthesized data on total P and total N concentrations in bulk precipitation and*

*throughfall from published literature to assess the status and characteristics of P deposition, N deposition, as discussed in detail below (section 2.2)".*

*In the revised section 2.1, we have included more detailed information on the sample collection and chemical analysis to make more clear that the method includes insoluble N and P by adding the following sentence: "
[revised manuscript text omitted]